# Improvement of Treg Selectivity and Stability for Diabetes Mellitus Type 1 Treatment: Complex Approach for Perspective Technologies

**DOI:** 10.3390/cells14221803

**Published:** 2025-11-17

**Authors:** Andrei A. Riabinin, Dmitry D. Zhdanov, Varvara G. Blinova, Alena A. Permyakova, Alina A. Stulova, Lyubov A. Rzhanova, Sofya Y. Nikitochkina, Elena I. Morgun, Ekaterina A. Vorotelyak

**Affiliations:** 1Koltzov Institute of Developmental Biology of Russian Academy of Sciences, 26 Vavilov Street, Moscow 119334, Russia; zhdanovdd@gmail.com (D.D.Z.); varya.blinova@list.ru (V.G.B.); dex.winner@gmail.com (A.A.P.); 9303923@gmail.com (L.A.R.); vorotelyak@yandex.ru (E.A.V.); 2Laboratory of Medical Biotechnology, Institute of Biomedical Chemistry, 10/8 Pogodinskaya Street, Moscow 119121, Russia; 3Department of Biochemistry, People’s Friendship University of Russia Named After Patrice Lumumba (RUDN University), 6 Miklukho-Maklaya Street, Moscow 117198, Russia; 4Biological Faculty, Lomonosov Moscow State University, Leninskiye Gory, 1, Moscow 119234, Russia; 5Department of Histology, Cytology and Embryology, People’s Friendship University of Russia Named After Patrice Lumumba (RUDN University), 6 Miklukho-Maklaya Street, Moscow 117198, Russia; stulovaalina976@gmail.com

**Keywords:** regulatory T cells, diabetes mellitus type 1, genetic modifications, epigenetic modifications, alternative splicing

## Abstract

**Highlights:**

**What are the main findings?**

**What are the implications of the main findings?**

**Abstract:**

The adoptive transfer of Tregs is a promising immunotherapeutic strategy for type 1 diabetes mellitus (T1D). A key focus in this field is the creation of antigen-specific CAR-Tregs targeted against pancreatic islet antigens. However, the efficacy of such therapies is potentially limited by the instability of the Treg phenotype in the inflammatory conditions of T1D. This review discusses molecular approaches to overcome this limitation. These include the genetic engineering of cytokine signaling pathways (IL2, IL33/ST2, and IL35) and the cAMP cascade, the management of FOXP3 splicing to ensure stable expression of concrete splice variants, and the use of epigenetic mechanisms to promote a durable Treg identity.

## 1. Introduction

Type 1 diabetes (T1D) is a chronic autoimmune disease mediated by T lymphocytes and characterized by the destruction of insulin-producing pancreatic beta cells. The etiology is multifactorial, resulting from a combination of genetic susceptibility and environmental triggers such as viral infections and toxins [1,2]. This pathological process is driven by autoreactive T cells, particularly CD8+ T lymphocytes, which infiltrate the islets and lead to a loss of insulin production [3].

Given this autoimmune basis, cell therapies utilizing regulatory T cells (Tregs) represent a promising therapeutic strategy. Tregs, defined as CD4+CD25+FOXP3+CD127low, are a specialized T-cell subset critical for maintaining immune homeostasis and peripheral tolerance by suppressing autoreactive T and B lymphocytes, as well as natural killer (NK) cells [4,5]. Potent immunosuppressive functions of Tregs make them ideal candidates for cell-based immunotherapy in autoimmune diseases like T1D [6].

Clinical trials investigating polyclonal regulatory T-cell (Treg) transplantation in both adult and pediatric T1D have consistently demonstrated the safety of this therapeutic approach. These studies have further reported a transient therapeutic effect, characterized by increased C-peptide levels and stabilization of HbA1c [7,8,9]. Early work by Marek-Trzonkowska et al. (2014) yielded promising results, suggesting that Treg administration could enhance the survival of pancreatic islets [8]. However, a subsequent follow-up study by the same group indicated that disease progression was not halted, with all patients reverting to insulin dependence within two years [7]. The collective data from these clinical trials have been summarized in a previous review [6]. The limited long-term efficacy observed with polyclonal Treg therapy may be attributable to the non-specific targeting of the pancreas by the infused cells.

To address this issue, researchers have created genetically engineered Tregs that express chimeric antigen receptors (CARs) specifically designed to target beta cells and improve their localization in the pancreatic islets. The chimeric antigen receptor (CAR) equips genetically modified Tregs with specificity and function through its two core domains: an antigen-binding domain that facilitates target recognition and adhesion, and a signaling domain that triggers cell activation. This design enables CAR-Tregs to home to their cognate antigen within damaged tissue and directly suppress the local inflammatory environment [6].

Studies evaluating these antigen-specific CAR-Tregs have yielded conflicting results.

The therapeutic potential of CAR-Tregs for T1D is critically dependent on both the careful selection of a target antigen and the optimal design of the CAR. Initial proof-of-concept studies highlighted these challenges. For instance, insulin-specific CAR-Tregs demonstrated strong antigen-specific proliferation and suppressive activity in vitro, yet were unable to prevent spontaneous diabetes in NOD (non-obese diabetic)/LtJ mice [10]. This failure was attributed to the inherent limitation of CARs being most effective against membrane-bound or oligomeric proteins, not soluble antigens like insulin [10,11].

This insight guided the next generation of CAR-Treg development. Spanier et al. hypothesized that for T1D therapy, CAR-Tregs must be activated in both the pancreatic islets and the pancreatic-draining lymph nodes. They therefore developed a CAR specific for a peptide from the insulin B chain (AA 10-23) presented in the context of the NOD mouse-specific Major Histocompatibility Complex II (MHCII) molecule, IAg7 [11]. This approach was highly successful; CAR-Tregs targeting this islet-specific pMHC demonstrated not only improved proliferation but also more pronounced suppressive activity, and completely prevented diabetes in NOD mice [11]. The success of targeting a tissue-relevant antigen is further supported by other work, such as that of Imam & Jaume, where GAD65-specific CAR-Tregs homed to pancreatic islets and reduced blood glucose levels in a humanized mouse model of T1D [12].

Conversely, the critical importance of stringent antigen selection is starkly illustrated by failed attempts at targeting the HPi2 epitope. HPi2-specific CAR-Tregs were initially activated but underwent a rapid functional collapse and numerical decline in vivo [13]. Subsequent investigation revealed that the monoclonal antibody used to construct the HPi2-CAR exhibited off-target binding to CD98, a ubiquitously expressed amino acid transporter found on all CD4+ T cells, including Tregs themselves [13]. This led to tonic, antigen-independent CAR signaling, resulting in activation-induced exhaustion and explaining the therapeutic failure [13].

Thus, the successful application of CAR-Treg therapy hinges on two principles: the selection of a target-restricted antigen to ensure precise targeting and the prevention of tonic signaling to avoid T cell exhaustion. These findings underscore that both antigen specificity and CAR design are paramount for translating this promising therapy to the clinic.

The results of these studies are summarized in Table 1.

A further complication for Treg therapies in T1D, even with antigen-specific CAR-Tregs, is Treg plasticity. This phenomenon, characterized by inflammation-induced phenotype switching, poses a substantial risk to therapeutic stability and efficacy [13].

A significant defect in Treg properties was observed in children with new-onset T1D, manifested as an expanded population of cells with low FOXP3 expression and intermediate CD25 expression, both of which are essential for Tregs functions and identity [14]. These cells were not suppressive and secreted pro-inflammatory IL17. It is still not clear whether these cells were activated effector T cells (Teffs) with transient expression of FOXP3, or if they represented a population committed from Tregs that failed to sustain FOXP3 expression [14]. Pancreas-infiltrating Tregs in NOD mice revealed enrichment of Tregs expressing the C-X-C chemokine receptor type 3 (CXCR3) [15]. Accumulation of CXCR3+ Tregs within pancreatic islets was dependent on the transcription factor T-BET, and genetic ablation of T-BET increased the onset and penetrance of disease [15]. Mice lacking T-BET+ Tregs showed a more aggressive insulitic infiltrate, reflected by elevated production of inflammatory cytokines [15]. Moreover, a subset of interferon-γ (IFN-γ)—producing CD4+CD25+FOXP3+CD127low Tregs has been observed in peripheral blood of patients with T1D [16]. Investigation of functional activity revealed that IFN-γ+ subset retained partial regulatory function, though with decreased potency relative to IFN-γ− Tregs. Furthermore, authors also conducted the epigenetic analysis of the Treg-specific demethylated region (TSDR) located on the FOXP3 gene. The hypomethylated status of TSDR is regarded to be a prerequisite for high expression of FOXP3 [17]. In IFN-γ+ Tregs of T1D patients, significantly lower levels of TSDR demethylation were observed compared to those in IFN-γ− Tregs. Moreover, in IFN-γ+ Tregs, the expression of the Ikaros family transcription factor Helios, associated with stable population of Tregs, was decreased [16]. Thus, the inflammatory microenvironment induces phenotypic changes in Tregs, leading to a deterioration in their therapeutic potential. One potential solution to this problem is to create Tregs that are resistant to the inflammatory microenvironment. This can be achieved through the use of cytokines and growth factors, as well as through genetic and epigenetic modification, and the control of FOXP3 alternative splicing. Here we discuss in detail these approaches.

## 2. Genetic Modifications for Tregs Stabilization Through Cytokine and cAMP Signaling Control

### 2.1. Interleukin 2 (IL2) Signaling

IL2 is a proinflammatory cytokine, and its signals influence various lymphocyte activities during differentiation, immune responses and homeostasis [18]. The phenotypic stability of cultured CD4+CD25+FOXP3+CD127low Tregs is critically sustained by IL2, in conjunction with TGF-β and CD3/CD28 stimulation [19]. Upon activation of the IL2/STAT5 signaling pathway in Tregs, phosphorylated STAT5 binds to the FOXP3 locus and stimulates its expression. The high-affinity interleukin-2 receptor (IL2R) is a trimeric complex composed of the alpha (CD25), beta (CD122), and common gamma (CD132) chains. The beta chain (CD122) is primarily responsible for ligand binding, while the gamma chain (CD132) facilitates intracellular signaling. The alpha chain (CD25) does not initiate signaling but dramatically increases ligand-binding affinity and complex stability. A functional, low-affinity receptor can be formed by the beta and gamma subunits alone; however, the incorporation of CD25 is essential for forming the high-affinity receptor that enables sustained IL2 signaling [20]. In humans and mice, the intermediate-affinity dimeric receptor (CD122/CD132) is constitutively expressed at low levels on naïve and memory CD4+ T cells, and at high levels on memory CD8+ T cells and CD56low NK cells. Also, IL2 can stimulate memory T cells [21], boost Teffs expansion [22] and IL2, stimulate dendric cells formation through innate and adaptive lymphoid cells in mice and humans [23]. Therefore, the administration of IL2 carries an inherent risk of non-selectively activating pro-inflammatory lymphocyte populations. However, it was shown that in the NOD mouse model, adenoviral-driven IL2 overexpression promoted the selective proliferation and activation of Tregs while avoiding the activation of Teffs, which ultimately conferred protection against T1D development [24]. Clinical studies demonstrated that administration of low-dose IL2 was safe and effective for the specific activation and expansion of Tregs in pediatric T1D patients [24]. However, a serious limitation is that endothelial cells express the high-affinity trimeric IL2R. Its activation by IL2 can induce Vascular Leak Syndrome, a known life-threatening side effect of high-dose IL2 therapy [25]. In lung endothelial cells, IL2R activation under exposure to a high dose of IL2 causes pulmonary edema [26]. Potential side effects of IL2 therapy are illustrated on Figure 1. Therefore, a more targeted approach would utilize genetically modified Tregs designed for inducible, specific activation of downstream IL2 effector functions. Genetic engineering of Tregs to achieve constitutive IL2 signaling, such as through the expression of membrane-bound IL2 (mbIL2) or a mutant orthogonal IL2 receptor (oIL2Rβ), represents a promising approach to enhance their therapeutic efficacy [27,28]. Supporting this, one study demonstrated a binary system where Tregs expressing a mutant oIL2Rβ were adoptively transferred into mice with graft-versus-host disease (GVHD) and treated with an orthogonal IL2 (oIL2) cytokine designed to bind exclusively to the engineered receptor. This strategy resulted in selective expansion of the modified Tregs in vivo, leading to improved survival and more effective suppression of pro-inflammatory CD4+ and CD8+ T cells in the intestine and secondary lymphoid organs [28]. An alternative, single-component strategy dispenses with an exogenous cytokine by engineering Tregs to express membrane-bound IL2 (mbIL2), which autonomously activates STAT5 signaling. This cell-intrinsic signaling stabilizes the Treg phenotype by enhancing FOXP3 transcription and stimulating IL10 production. In a xenogeneic GVHD model using NRG (Non rag gamma: NOD, Rag1 and IL2 knockdown) mice, these mbIL-2-expressing CAR-Tregs demonstrated a significant survival advantage over control CAR-Tregs, confirming the functional benefit of autonomous signaling [27].

### 2.2. Interleukin 33 (IL33) Signaling

Beyond the critical role of IL2, the stabilization and function of Tregs are modulated by other key cytokines, such as IL33. A member of the IL1 family, IL33 is a crucial regulator of T cell differentiation, function, and immune homeostasis [29]. IL33 can specifically enhance the anti-autoimmune activity of Tregs in the context of T1D. In a streptozotocin-induced T1D model, IL33 administration prevented disease onset by promoting the proliferation of ST2+FOXP3+ Tregs in pancreatic islets and draining lymph nodes [30]. Ex vivo studies using Tregs from diabetic patients have shown that IL33 stimulation significantly expands the CD4+CD25highFOXP3+ population and enhances their suppressive capacity, particularly in inhibiting IFN-γ production by Teffs [30,31]. The correlation between FOXP3 and its receptor ST2 in diabetic Tregs upon IL33 stimulation suggests that FOXP3 expression may be regulated through the ST2 signaling pathway [32].

The ST2+ Treg cell subset is a population characterized by a highly activated state, even under homeostatic conditions in mice. This subset is present across various non-lymphoid tissues, including the lungs, intestines, skin, adipose tissue, liver, and muscles [33,34,35,36,37]. Functionally, murine ST2+ Tregs demonstrate superior suppressive capacity compared to their ST2- counterparts in vitro, irrespective of IL33 signaling [38]. This enhanced immunosuppressive activity is partly attributed to their increased production and activation of the anti-inflammatory cytokines IL10 and transforming growth factor beta (TGF-β). Thus, ST2 expression serves as a marker for a highly activated and potently suppressive Treg population that predominantly resides within non-lymphoid organs.

In contrast, data on ST2 expression in human Tregs are limited and contradictory. Some studies report an absence of ST2+ Tregs in human blood, tonsils, healthy colon, synovial fluid from juvenile idiopathic arthritis patients, and lung tissue post-transplantation [39]. Conversely, other research has identified this population in the blood and synovial fluid of rheumatoid arthritis patients [40], and high ST2 expression has been observed on Tregs within the human omentum [35]. These discrepancies highlight the need for further investigation into the presence and role of ST2+ Tregs in humans.

To address this, a recent study generated ST2+ human Tregs through lentiviral transduction. These engineered ST2+ Tregs exhibited enhanced TCR-dependent proliferation and expansion upon exposure to IL33, alongside increased expression of the proliferation marker Ki-67 [39]. Furthermore, expansion in the presence of IL33 significantly upregulated key Treg-associated proteins, including CD25, FOXP3, CTLA-4, and LAP, while maintaining high expression of the stability marker Helios. IL33 also enhanced TCR activation, as evidenced by increased CD39, CD71, and HLA-DR expression, suggesting increased resilience of the transduced cells [39].

However, a potential concern is that ST2+ Tregs can exhibit a Th2-like character, expressing the transcription factor GATA-3 and producing the cytokines IL5 and IL13, particularly in response to IL33 [38], raising the possibility of phenotypic instability in human cells [39]. Despite this, a therapeutically promising finding is that these engineered ST2+ Tregs promoted the alternative activation (M2 polarization) of allogeneic CD14+ monocytes in co-culture, indicated by upregulated CD163 and CD206 expression—an effect beneficial for treating conditions like T1D [39]. Crucially, this effect was independent of IL33 stimulation, providing a major therapeutic benefit by circumventing the need for systemic IL33, which has dual drawbacks: it exerts non-selective pro-inflammatory effects on various immune cells, including CD4+ and CD8+ T cells and mast cells, and can directly promote tumor progression by activating cancer cell proliferation, survival, and metastasis [41,42,43,44].

Collectively, the generation of genetically engineered ST2+ Tregs represents a promising therapeutic strategy, but one that necessitates careful further investigation to fully understand its potential and limitations.

### 2.3. Interleukin 35 (IL35) Signaling

IL35 is an anti-inflammatory cytokine from the IL12 family. It is produced by Tregs and plays a role in immune suppression by blocking the development of Th1 and Th17 cells through limiting early T cell proliferation [45,46]. The functional IL35 receptor (IL35R) consists of various dimeric configurations, encompassing homodimers of IL12Rβ2, homodimers of gp130, or heterodimers composed of both subunits [47]. Huang et al., showed that when exposed to IL35, phosphorylation of the transcription factor STAT1, which stabilizes FOXP3 expression, occurs via gp130 activation [48]. It can be assumed that STAT1 activation occurs under the influence of JAK [49]. There is another mechanism for stabilizing the Treg phenotype through IL35. In a murine model of streptozotocin-induced T1D, IL35 administration stabilized the Treg phenotype by upregulating the expression of Eos, a critical transcription factor for Treg lineage stability [50]. Eos in combination with FOXP3 provides the repression of a number of genes encoding T-effector cytokines, such as IL2 and IFN-γ. Thus, it is possible that the stabilization of Treg using IL35 occurs through a dual mechanism—directly through the activation of FOXP3, as well as through the complex interaction of Eos and FOXP3.

This signaling promoted a robust Treg response, characterized by increased expression of canonical markers FOXP3 and PD1, a central memory phenotype CD44+CD62L-, and proliferation marker Ki-67. Furthermore, IL35 exposure induces autocrine production of IL35 in Tregs, establishing a positive feedback loop that amplifies its own expression. This was associated in vitro by a dramatic increase in IL35 concentration and a concurrent decrease in pro-inflammatory IL17A in culture supernatants, confirming that IL35 supports Treg stability, expansion, and suppressor function [48]. However direct IL35 therapy has some potential side effects. IL35 causes hepatocytes proliferation and liver regeneration decrease [51], as well as stimulates tumor growth [52,53] (Figure 1). As an alternative, targeting the IL35 pathway through Treg-specific genetic engineering can avoid the side effects of cytokine therapy and remains an unexplored therapeutic avenue. Such an approach could confer the benefits of IL35 signaling-enhanced Treg stability, while circumventing the off-target effects on other T cell populations associated with cytokine therapy. However, no such studies currently exist. Directly translating engineering strategies from the IL2 pathway to IL35 presents significant challenges. Approaches involving mutant ligand-receptor pairs or membrane-bound proteins may not be directly applicable. This is due to the complex, heterodimeric nature of the IL35R, whose activation mechanisms remain poorly characterized. Furthermore, the IL35 ligand itself is a heterodimer of IL12α and EBI3 (IL27β), unlike the single-chain IL2 [47]. Consequently, a comprehensive understanding of the IL35/IL35R signaling mechanism in Tregs is a prerequisite for its successful engineering, which will likely necessitate novel, distinct strategies.

### 2.4. Cyclic AMP (cAMP) Signaling

Treg stability can be enhanced not only by modulating interleukin signaling but also by targeting intracellular metabolic pathways such as cAMP signaling. Evidence from genetically modified mice overexpressing microRNA-142-5p demonstrates that reduced expression of phosphodiesterase 3B (Pde3b), which degrades cAMP augments the anti-autoimmune activity of Tregs. Conversely, Pde3b overexpression leads to early mortality, underscoring the critical role of cAMP levels in Treg function [54]. This pathway is intrinsically linked to the Treg lineage-defining transcription factor FOXP3. High FOXP3 expression correlates with reduced PDE3B levels [55] and increased activity of adenylate cyclase 9 (AC-9), which catalyzes cAMP production [56]. Blockade of cAMP degradation by PDE inhibition improves Treg-mediated suppression in a murine asthma model [57]. The transcription factor CREB (cAMP response element-binding protein) serves as a critical molecular link between cAMP signaling and FOXP3 expression in Tregs. Phosphorylated CREB acts as a direct transcriptional activator of the FOXP3 gene, thereby promoting the Treg lineage [58]. Also CREB modulates Treg phenotype and function by restricting chromatin accessibility at the ST2 locus, which encodes the receptor for IL33. This limitation of ST2 availability fine-tunes the IL33 signaling pathway, a key mechanism for Treg stabilization in inflammatory and tissue contexts, ultimately shaping the resulting immune response [59].

Collectively, these findings indicate that FOXP3 expression correlates with a high intracellular cAMP environment, which in turn contributes to the stabilization of the suppressive Treg phenotype.

Thus, genetically engineered induction of the IL2, IL33, and IL35 signaling pathways, alongside cAMP signaling stimulation, can stabilize the Treg phenotype and enhance suppressive function. Consequently, the development of novel genetic strategies to enforce Treg stability represents a critical research priority, as summarized in Table 2.

## 3. Modulation of Treg Stability, Suppressive and Proliferative Activity Through FOXP3 Alternative Splicing

The Treg phenotype is critically dependent on the expression of FOXP3, which serves as their primary lineage-defining marker. This transcription factor is indispensable for regulating Treg differentiation, stability, proliferation, and suppressive activity [4]. Functionally, FOXP3 belongs to a large, evolutionarily conserved family of Forkhead transcription factors.

These proteins interact with specific target sites, known as splicing enhancers or silencers on the pre-mRNA molecule [3]. The structure of FOXP3 consists of N-terminus repression region (epigenetic and transcription control), zinc-finger DNA-binding domain, leucine zipper domain (FOXP1 interaction) and forkhead DNA-binding domain [61] (Figure 2A). The human FOXP3 gene consists of 12 exons (Figure 2B), 11 of which encode the full-length protein (FOXP3FL). Alternative splicing of FOXP3 pre-mRNA results in the formation of three more alternatively spliced variants (Figure 2C): a variant with a deletion of exon 2 (FOXP3Δ2); a variant with a deletion of exon 7 (FOXP3Δ7); and a splice variant with deletions of both exons 2 and 7 (FOXP3Δ7) [62,63].

Deletion of exon 2 impacts a portion of the proline-rich repressor region, leading to changes in the ability of FOXP3 ∆2 and ∆2∆7 variants to interact with partner proteins. With the deletion of exon 7, a segment of the leucine zipper domain is lost, causing disruption in the formation of FOXP3 dimers by FOXP3 ∆7 and ∆2∆7 variants. Furthermore, nuclear export signals are encoded by the boundaries of exons 1–2 and 6–7 [64]. Deletion of exons 2 and/or 7 results in the absence of nuclear export signals in FOXP3Δ2, FOXP3Δ7, and FOXP3Δ2Δ7 (Figure 2B), leading to differences in the subcellular localization of the splice variant protein forms [64]. FOXP3Δ2 and FOXP3Δ7 proteins exhibit a greater tendency toward nuclear localization compared to FOXP3FL, while FOXP3Δ2Δ7 is almost entirely localized to the nucleus. Thus, the ability to interact with partner proteins and different subcellular localization influences the transcriptional activity of different FOXP3 splice variants and, consequently, Treg biology.

Since Tregs play a crucial role in regulating and maintaining peripheral immunological tolerance, there is active research on the relationship between FOXP3 splice variants and the function of these cells in autoimmune diseases. An intriguing finding was the detection of mRNA for all four splice variants in CD4+ T lymphocytes from healthy donors and the proportion of FOXP3FL increased during in vitro transformation into Treg over a period of 9 days [65]. Transformed Tregs exhibited stronger suppressive activity, indicating that FoxP3FL may play a role in induction the suppressive activity of Tregs.

Two studies have shown that the proportion of FOXP3 splice variants in Tregs differs between patients with multiple sclerosis [66] or amyotrophic lateral sclerosis [67] and healthy donors. In healthy donors, FOXP3FL and FOXP3∆2 variants are predominant, while in patients with aforementioned autoimmune diseases, FOXP3∆7 and FOXP3∆2∆7 are more common. The decrease in the proportion of FOXP3FL in Tregs of patients with autoimmune diseases is consistent with the decrease in the number of Tregs and their suppressive activity in peripheral blood in these patients. T1D is one of the manifestations of IPEX syndrome, which is characterized by dysfunctional Tregs in which the expression of FOXP3∆2 dominates over that of FOXP3FL. According to Jianguang Du et al., this is due to a mutation in the second exon of the FOXP3 gene (305delT) [68]. It has also been demonstrated that butyrate (an epigenetic factor and histone deacetylase inhibitor) and IFN-γ (an immunostimulatory cytokine) can alter the FOXP3∆E2:FOXP3FL ratio in RVMCs. This suggests that in autoimmune diseases like T1D, the altered ratio of these isoforms may stem not only from genetic predisposition but also from non-heritable, inflammatory factors within the tissue microenvironment [69].

In order to selectively induce the expression of individual FOXP3 splice variants and study their impact on Treg biology, an approach was taken to modulate the FOXP3 alternative splicing with splice-switching oligonucleotides (SSOs). SSOs are antisense oligonucleotides that target splicing modulation [70].

It has been shown that the FOXP3 pre-mRNA in the regions of exon 2 and exon 7 contains binding sites for splicing-regulating SR proteins, which are responsible for the insertion or deletion of exons.

According to one study [66], 36-mer oligonucleotides have been designed to base-pair with FOXP3 pre-mRNA in regions sensitive to regulatory splicing proteins, blocking their ability to regulate exon skipping processes. The authors claimed that the designed SSOs do not have off-target regions across the human genome. The delivery of the designed SSOs to Tregs was performed by in vitro transfection with the liposomal agent Lipofectamine 2000.

Transfecting of healthy donor Tregs with pairs of specific SSOs allowed directing the alternative splicing process towards selective synthesizing only one splice variant and obtaining Tregs lines with selective expression of one of the four FOXP3 splice variants. Induction of FOXP3FL was accompanied by a 1.5- to 2-fold increase in the proliferative activity of these cells compared with control cells. Additionally, cells with FOXP3FL had a 2-fold greater ability to suppress effector autologous lymphocytes compared to control cell line. The decrease in suppressor activity was linked to a reduced ability of these cells to synthesize molecules involved in the suppressive action of Tregs [66,67].

The ability of SSOs to selectively induce FOXP3FL and produce Tregs with enhanced proliferative and suppressor activity has enabled the use of these oligonucleotides for regenerative therapy [66]. Tregs were isolated from the peripheral blood of patients with amyotrophic lateral sclerosis. Transfection of Tregs with a pair of SSOs specific for splicing-regulating sequences on exon 2 and exon 7 of FOXP3 pre-mRNA allowed for redirection of alternate splicing towards FOXP3FL synthesis. Tregs with this splice variant underwent significantly faster division than control cells from the same patients. The induction of FOXP3FL was associated with an increased ability of these cells to synthesize suppressor molecules and inhibit telomerase. As a result, the obtained cells were able to suppress target effector lymphocytes more actively compared to control cells. The authors claimed that the developed technology is applicable for regenerative therapy of the suppressor link of immunity in patients with amyotrophic lateral sclerosis. Although this technology has not yet been tested for T1D-Tregs, it appears to be a potentially powerful tool for generating FOXP3FL expression in cells for the regenerative treatment of T1D. Thus, understanding the mechanisms that regulate alternative FOXP3 splicing and exploiting the chemical factors that control this process may enable Treg activity to be modulated in vitro and in situ, providing an alternative therapy for a number of autoimmune diseases, including T1D.

## 4. Treg Stabilization by Epigenetic Modification

Epigenetic modification represents a promising strategy for establishing a stable Treg phenotype. Key regulatory elements upstream of the Foxp3 locus include three conserved non-coding sequences (CNS1-3) [71]. Among these, CNS2, the Treg-specific demethylated region (TSDR) is critical for stable Foxp3 expression. Demethylation of the TSDR promotes a euchromatic state that enables the persistent binding of transcription factors like STAT5 and Ets1, creating a positive feedback loop to lock in Foxp3 expression [72,73]. This epigenetic fortification is crucial for Treg stability, as it confers resistance to inflammatory cytokines (e.g., IL6, IFN-γ) that would otherwise drive pathogenic conversion to IFN-γ-producing effector phenotypes, thereby protecting Treg identity [74,75]. The integrity of this mechanism is thus a primary indicator of Treg function.

The pathophysiological relevance of TSDR demethylation is starkly evident in autoimmune conditions like T1D. The relentless inflammatory microenvironment of the pancreatic islets can actively induce hypermethylation of the TSDR, leading to a loss of Treg identity and function [76]. This is corroborated by clinical observations showing that patients with T1D have an increased frequency of unstable FOXP3+IFN-γ+ Tregs, which specifically lack the characteristic TSDR demethylation. This epigenetic defect directly correlates with reduced transcriptional stability of FOXP3 and diminished suppressive capacity [16]. Conversely, therapeutic intervention in pre-clinical models using the demethylating agent 5-Aza-2′-deoxycytidine (DAC) in NOD mice prevented diabetes onset by preserving TSDR demethylation, thereby enhancing Treg stability and function. This confirms the direct therapeutic potential of targeting this specific epigenetic pathway [77].

Beyond DNA methylation, the FOXP3 locus is further stabilized by a repertoire of histone modifications. Activating marks such as H3K27 acetylation (H3K27ac), which opens chromatin, and H3K4 monomethylation (H3K4me1), a mark of active enhancers, at the FOXP3 locus promote the euchromatic state and facilitate the recruitment of transcriptional co-activators [78]. These modifications work in concert with TSDR demethylation to create a resilient, self-reinforcing epigenetic landscape that locks in the Treg gene expression program, a complexity explored in further detail in related studies [79].

A diverse array of methods can be employed to deliberately modify the epigenome of Tregs, thereby reinforcing their stability. These approaches range from pharmacological interventions—such as DNMT [80] and HDAC inhibitors [81] to physiological strategies like the activation of TET enzymes via vitamin C [82] or hypothermia [83]. Further methods include modulating signaling pathways by reducing CD28 stimulation or inhibiting cyclin-dependent kinases, alongside the highly precise technique of CRISPR-dCas9 epigenetic editing [84]. Collectively, these tools enable the creation of “epigenetically fortified” Tregs for adoptive cell therapy. Such engineered cells are designed to resist inflammatory conversion and provide durable immune regulation in the context of autoimmune diseases, transplantation, and other inflammatory disorders, including T1D.

## 5. Complex Approach

The therapeutic promise of Tregs for establishing a durable immunosuppressive niche in the islets of Langerhans is poised to be unlocked through multi-layered engineering. Overcoming the fundamental challenge of Treg instability in the inflammatory milieu of T1D requires a synergistic strategy that moves beyond singular approaches, which are summarized in Figure 3. The autologous Tregs extracted from patient blood can be cultivated and analyzed for further modification optimization. The next step is autologous CAR T-reg generation (DAG65R/HPiR/insulin receptor+) for target homing (beta cells zone). The generated cell lines can undergo the next stage of modifications aimed at obtaining stabilized CAR Tregs by epigenetic modification (acetylation of H3K27 and monometylation of H3K4, demethylation in TSDR in FOXP3 region), genetic modification (aimed in IL2/IL33/IL35/cAMP signaling upregulation) or FOXP3FL isoform alternative splicing switching by SSOs. While combining multiple methods for Treg stabilization holds the potential for superior efficacy compared to single modifications, this approach presents significant challenges. The increased complexity increases manufacturing costs and raises potential safety concerns, such as unintended Treg differentiation or the induction of cell death. These risks necessitate further investigation.

From a therapeutic standpoint, a comprehensive platform capable of evaluating diverse modifications to enhance the selectivity and potency of patient-specific Tregs represents a theoretically flexible strategy for T1D. This is critical given the inherent variability in treatment response across Treg cell lines derived from different donors.

A major translational hurdle, however, is the regulatory pathway. The registration of Advanced Therapy Medicinal Products (ATMPs) based on multi-modified Tregs is a substantially longer and more resource-intensive process compared to mono-modified products, as it requires a more extensive and complex series of clinical trials [85].

The identification of the safest, most stable, and most effective autologous CAR-stabilized Treg cell line requires a comprehensive validation strategy. This involves a multi-parametric testing battery to rigorously characterize the product. Core assessments include immunophenotyping by flow cytometry to confirm a canonical Treg profile (CD4+, CD25+, FOXP3+, CD127low) [86] and a functional Treg suppression assay to quantify the capacity to inhibit the proliferation of autologous conventional T cells (Tconv) in co-culture [87].

Further characterization entails evaluating cellular survivability by assessing the ratio of proliferation to apoptosis, using markers such as Annexin V for apoptosis quantification [88], BrdU incorporation assays and Ki67 detection for proliferation analysis by flow cytometry [89]. A critical functional attribute is the cytokine profile, which should demonstrate a pro-tolerogenic signature—characterized by the production of TGF-β, IL2, and IL10 and an absence of pro-inflammatory cytokines like IFN-γ, IL17A, IL6, and TNF-α—a profile that can be corroborated at the transcriptional level via sequencing [90,91]. From a safety perspective, karyotyping is indispensable for screening chromosomal abnormalities that could indicate oncogenic potential [92].

Finally, the most representative functional validation is achieved through a three-component in vitro system. This model co-cultures pancreatic beta cells with inflammatory effector cells to induce cytotoxicity, and it engineered stable CAR-Tregs as a protective agent. The therapeutic efficacy is then quantified by the stable CAR-Tregs’ ability to preserve beta-cell viability, which is measured using metabolic assays such as MTT or resazurin, which detect the activity of NAD(P)H-dependent oxidoreductase enzymes in living cells via a change in optical density [93].

Following successful validation, the qualified cell line is expanded to a therapeutically relevant dose and subsequently administered to the patient via adoptive cell transfer.

Patient-derived Tregs are first isolated and expanded ex vivo. They subsequently undergo two key engineering steps: (1) Targeting: Genetic modification to express a chimeric antigen receptor (CAR) specific for islet-associated antigens (e.g., GAD65 or insulin B chain) to enable homing to pancreatic islets. (2) Stabilization: Enhancement of Treg phenotype stability and function through epigenetic modulation (e.g., DNMT/HDAC inhibition, CRISPR-dCas9 systems), alternative splicing of FOXP3 using splice-switching oligonucleotides (SSOs), or genetic engineering to promote anti-inflammatory cytokine production, survival, and proliferation. The resulting stabilized CAR-Treg cell line is rigorously analyzed for safety and efficacy before being expanded and injected into the patient.

## 6. The Application of Good Manufacturing Practice (GMP) in Genetically Modified Treg Therapy

Good Manufacturing Practice (GMP) is fundamental to the manufacturing of cell-based therapies, including genetically engineered Tregs. GMP guidelines ensure consistent production, rigorous quality control, and the overall safety and efficacy of the final product. The implementation of GMP revolves around three critical pillars: comprehensive product safety control, strict adherence to established standards, and the cultivation of trust among stakeholders.

### 6.1. Product Safety and Genomic Integrity

Ensuring product safety is paramount. The generation of stabilized CAR-Tregs involves sequential genetic modifications, and permanent alterations to the Treg genome necessitate extensive testing to detect unintended consequences. These include off-target genomic effects such as frameshift mutations, gene downregulation, or other inadvertent genetic alterations that could potentially promote tumorigenesis. Therefore, the genomic and phenotypic stability of modified Tregs must be rigorously validated using a combination of single-cell genome, transcriptome, and epigenome sequencing, proteomic analysis, and flow cytometry. Additionally, the final cell product must be verified as free from pathogenic contamination, a requirement typically fulfilled through PCR-based testing [94].

### 6.2. Compliance with GMP Standards

Demonstrating compliance with GMP standards reflects a commitment to quality and safety. Immune cells for therapeutic use must be manufactured via standardized, reproducible, and GMP-conformant processes. A significant challenge is the development of a robust and unified manufacturing protocol for modified Tregs to facilitate their entry into clinical trials. Achieving consistent and reproducible cell products demands meticulous control and documentation of all process parameters throughout the cultivation period. This includes stringent oversight of environmental conditions such as temperature (37 °C), pH (7.2–7.4), oxygen (1–21%), carbon dioxide (5%), and humidity (90–95%), as well as the concentrations of critical media components like glucose, lactate, amino acids, ions, activation stimuli (e.g., anti-CD3/CD28 beads), and growth factors (e.g., IL2, TGF-β) [95]. Cell culture must utilize a uniform, certified basal medium, such as DMEM or RPMI [95]. Furthermore, manufacturing steps should ideally be performed in a Clean Room class A environment, and the final product configuration should be an automated process to minimize batch-effects [96].

The initial Treg population must be phenotypically defined (CD4+, CD25+, and FOXP3+) [86], and the therapeutic potency of the final product must be demonstrated through functional assays, such as a suppression assay using autologous Tconv. Compliance with national and international regulatory requirements is mandatory for approval as an investigational medicinal product (IMP) or market authorization as an Advanced Therapy Medicinal Product (ATMP), for which autologous Tregs are typically required. Adherence to GMP ensures the stability, reproducibility, and effectiveness of the Treg-based IMP [97]. Comprehensive documentation is essential and must include (a) the complete Treg cell line history; (b) all relevant development data; (c) specifications for all materials and equipment; and (d) all procedures as Standard Operating Procedures (SOPs), alongside a full set of analytical data for the final product [96].

### 6.3. Building Trust Through Rigorous Practices

Adherence to GMP standards is crucial for building confidence among patients, clinicians, and investors. The use of well-validated, reliable technologies and high-quality materials for cell extraction, genetic modification, cultivation, and analysis is non-negotiable. Key strategies to enhance the safety profile and credibility of the ATMP include avoiding serum during Treg cultivation [98] and minimizing the use of pathogenic viral vectors, such as those based on lentiviruses, coxsackievirus, rotavirus, or poliovirus, for genetic modifications [99,100].

In summary, the successful clinical translation of genetically modified Treg therapies is inextricably linked to rigorous GMP implementation. A holistic approach that integrates stringent safety testing, controlled and documented manufacturing processes, and the use of safe, validated materials is indispensable for ensuring product quality, securing regulatory approval, and ultimately, establishing the trust required for therapeutic application.

## 7. Conclusions

The integration of antigen-specific homing via CARs with intrinsic phenotypic fortification represents a paradigm shift for T1D treatment. By genetically reinforcing key stability pathways such as IL2, IL35, and IL33/ST2, as well as cAMP signaling; directing FOXP3 splicing towards the fully functional FOXP3FL isoform; and enforcing a stable epigenetic landscape at the FOXP3 locus, we can create a new generation of “armored” Tregs. These modified Tregs are designed to precisely target the site of autoimmunity, maintain their regulatory identity under inflammatory pressure, and potently suppress the immune attack. This holistic integration of targeting genetic and epigenetic fortification offers a robust and compelling path forward to durably halt autoimmune beta cell destruction and achieve lasting immune tolerance.

## Figures and Tables

**Figure 1 cells-14-01803-f001:**
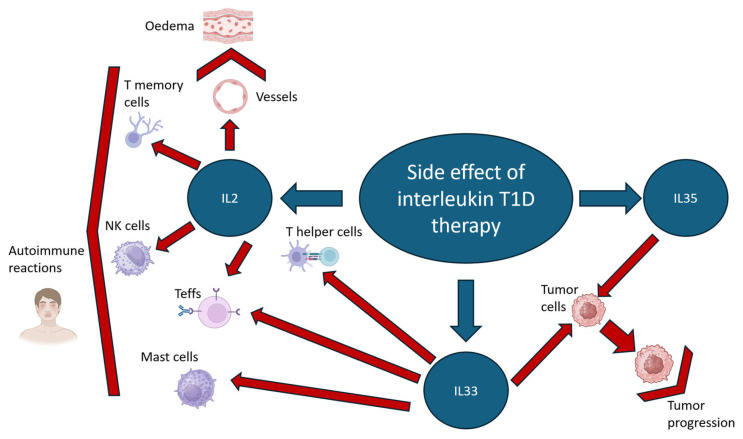
Potential side effects of IL2, IL33 and IL35 therapy based on direct injection. IL2 and IL33 T1D therapy can induce autoimmune reactions by paracrine effects on endothelial cells, NK cells, Teffs, T helper cells and mast cells. IL33 and IL35 therapy can provoke tumour progression. Red arrow indicates target cells for IL2, IL33 and IL35 , bracket-shaped arrows indicate the effects of these interleukins. Created in BioRender. Riabinin, A. (2025) https://BioRender.com/d2pqwqg (accessed on 10 November 2025).

**Figure 2 cells-14-01803-f002:**
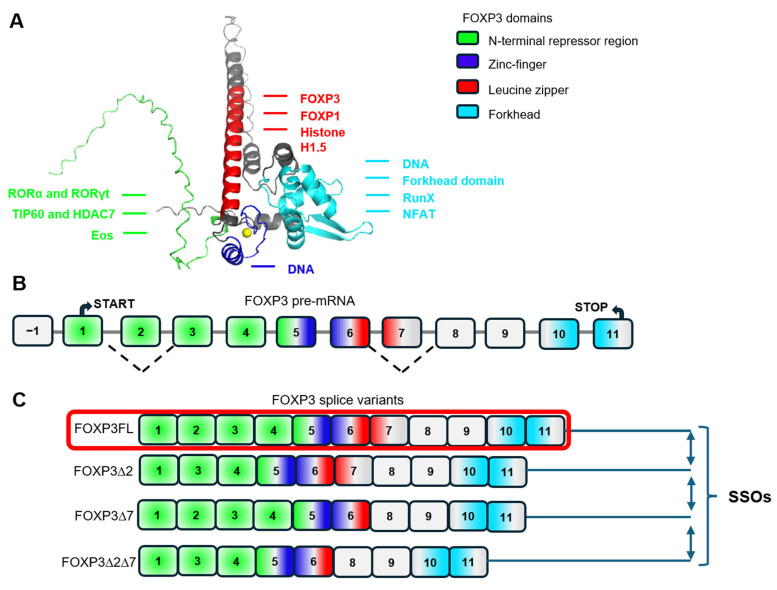
Domain structure of human FOXP3 and its splice variants. (**A**) Domain structure of human FOXP3 protein its partner molecules. The domains are colored according to the legend. The Zn2+ ion is shown by a yellow ball. Partner molecules are colored according to the interacting domain. ROR—RAR related orphan receptor; TIP60—histone acetyltransferase; HDAC—histone deacetylase; EOS—transcription cofactor; RunX—Runt-related transcription factor 1; NFAT—Nuclear factor of activated T-cells. The structure was generated in AlphaFold3 and visualized in PyMOL 2.5.2. (**B**) The schematic presentation of FOXP3 pre-mRNA consisting of one non-coding (−1) and 11 coding (1–11) exons. The open reading frame, limited by start-, and stop-codons, is shown. Exons 2 and 7 are subjected to alternative splicing, as indicated by dotted line. Dashed brackets indicate which exons are juncted during alternative splicing (**C**) The schematic presentation of four existing FOXP3 splice variants that result from pre-mRNA alternative splicing. In panels (**B**,**C**), exons are colored according to the FOXP3 domains they encode, as indicated in the legend. The corresponding box color code is identical with FOXP3 domains color code (**A**). The FOXP3 spliceform (FOXP3FL), the most efficient in terms of maintaining a stable Treg phenotype, is circled in red. The alternative splicing of FOXP3 can be controlled by SSOs.

**Figure 3 cells-14-01803-f003:**
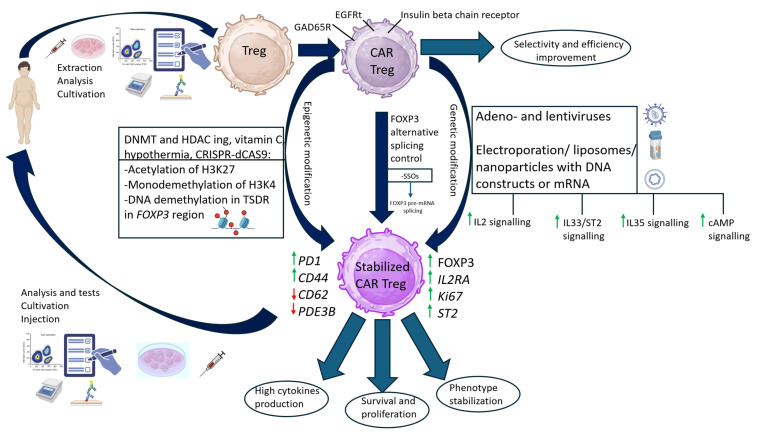
Strategy for Engineering Stable and Targeted CAR-Tregs for T1DCreated in BioRender. Riabinin, A. (2025) https://BioRender.com/icps73f (accessed on 10 November 2025).

**Table 1 cells-14-01803-t001:** Targets for CAR-Treg T1D treatment.

Epitope/Antigen	Study Date	Mouse Strain	Model In Vivo	Results/Efficiency	Reference
Insulin	2019	NOD/LtJ	Spontaneous autoimmune diabetes	Insulin-specific CAR-Tregs were functional in vitro but failed to prevent diabetes in NOD/LtJ mice.	Tenspolde et al., 2019 [10]
HPi2 (pancreatic marker)	2020	-	-	HPi2-specific CAR-Tregs failed due to off-target CD98 binding and consequent exhaustion	Radichev et al., 2020 [13]
EGFRt	2024	NSG	Graft rejection was modeled in NSG mice by challenging established EGFRt-sBC transplants with an adoptive immune transfer of CAR-Teffs ± CAR-Tregs.	EGFRt-specific CAR-Tregs, generated against an engineered inert target on hPSCs, demonstrated potent suppression of innate and adaptive immune responses in vitro and completely prevented the immune rejection of stem cell-derived pancreatic beta-like cell grafts in vivo.	Barra et al., 2024 [14]
Insulin beta chain (AA 10-23)	2023	NOD	Spontaneous autoimmune diabetes Diabetes induced in immunodeficient NOD mice by BDC2.5 T cell transfer	CAR-Treg therapy completely prevented diabetes in both models, showing stability and a potent suppressive effect.	Spanier et al., 2023 [11]

**Table 2 cells-14-01803-t002:** Perspective Genetic Engineering Approaches for Treg Phenotype Stabilization.

Signaling Pathway	Methodology	The Result of Therapy	References
IL2	Overexpression of IL2 by adenovirus	Prevention T1D development in NOD mice	Churlaud et al., 2014 [24]
	Creation of a vaccine based on autologous T-regs after culturing in the presence of IL2, antibodies to CD3/CD28, TGF-β	The rate of T1D exacerbations more than halved, and the EDSS score increased by about 10%.	Eliseeva et al., 2016 [19]
	Introduction of a plasmid expressing proinsulin 2 and a combination of immunomodulatory cytokines (transforming growth factor-β1, interleukin IL10 and IL2).	Reduction in the incidence of T1D development in the NOD line of mice prone to this disease to 0	Pagni et al., 2022 [60]
IL33	Direct injection of IL33	Prevention of T1D development in lymph nodes and pancreatic islets in a streptoztocin-induced T1D model through increased ST2+Foxp3+ Treg proliferation	Pavlovic et al., 2018 [30]
Il35	Direct injection of exogenous IL35 into c T1D mice	T-reg stabilization	Singh et al., 2015 [50]

## Data Availability

No new data were created or analyzed in this study.

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
