# Peer review of "Improvement of Treg Selectivity and Stability for Diabetes Mellitus Type 1 Treatment: Complex Approach for Perspective Technologies"

_cells, 2025, doi:10.3390/cells14221803_

Round 1

Reviewer 1 Report

Comments and Suggestions for Authors

This manuscript presents a timely and comprehensive mini-review on a critically important topic in Treg-based immunotherapy for Type 1 Diabetes (T1D). The authors effectively synthesize a substantial body of literature to argue that overcoming Treg instability within the inflammatory milieu of T1D represents a key therapeutic challenge. They propose a "complex approach" that integrates antigen-specific targeting with intrinsic stabilization strategies. The focus on genetic engineering approaches, particularly cytokine signaling and cAMP modulation, alongside FOXP3 splicing and epigenetic modifications, is well justified and aligns with current advancements in the field. The review is generally well-structured, and the figures and tables effectively summarize complex information. Nevertheless, the manuscript in its current form would benefit from revisions aimed at enhancing clarity and deepening the critical analysis in several sections. Specific comments are as follows.

  1. In the section of “Genetic modifications for Tregs stabilization throw cytokine and cAMP signaling control”, the discussion of the orthogonal IL-2 system and mbIL-2 is excellent. However, for IL-33 and IL-35, the review mostly focuses on the effects of cytokine administration rather than the engineering of Tregs to exploit these pathways, which is the core theme. This section should be reframed to explicitly discuss the potential and challenges of, for example, engineering Tregs to constitutively express ST2 or the IL-35 receptor, and what the expected functional outcomes and potential risks might be.
  2. Certain sections lack sufficient depth, with relatively general descriptions of specific mechanisms, such as the precise role of the IL-35 signaling pathway in Tregs and the direct link between cAMP signaling and FOXP3 expression. These aspects warrant further elaboration. It is recommended to enhance the key mechanistic discussions by incorporating more detailed molecular-level insights, including the interactions among critical signaling proteins and the transcriptional regulatory networks governing FOXP3 expression.
  3. In the section of “Complex Approach” is a key section but is currently underdeveloped. The proposed pipeline is logical but highly generic. The authors should significantly expand and add specificity to "Complex Approach" with testable hypotheses and a discussion of practical challenges.
  • How would one prioritizewhich stabilization method(s) to use? Would a combination be synergistic or redundant?
  • Discuss the practical and regulatory hurdles of implementing such a multi-layered engineering approach.
  • "The most secure, stable and effective stabilized autologous CAR stabilized Treg cell line can be selected after a series of tests". What specific in vitroand in vivo assays would be used for this selection? The authors mention stability tests; they should specify metrics.
  1. Perform a careful proofread of all figures and captions. “Potential side effects of IL2 therapy are illustrated on Fig. 2”, the figure is wrongly matched, this content is displayed in Fig. 1 rather than Fig. 2.
  2. Authors are kindly reminded to pay attention to formatting details; for instance, terms such as "in vivo" and "in vitro" should be typeset in italics in accordance with standard scientific writing conventions.

Author Response

Answers to reviewer 1 comments:

This manuscript presents a timely and comprehensive mini-review on a critically important topic in Treg-based immunotherapy for Type 1 Diabetes (T1D). The authors effectively synthesize a substantial body of literature to argue that overcoming Treg instability within the inflammatory milieu of T1D represents a key therapeutic challenge. They propose a "complex approach" that integrates antigen-specific targeting with intrinsic stabilization strategies. The focus on genetic engineering approaches, particularly cytokine signaling and cAMP modulation, alongside FOXP3 splicing and epigenetic modifications, is well justified and aligns with current advancements in the field. The review is generally well-structured, and the figures and tables effectively summarize complex information. Nevertheless, the manuscript in its current form would benefit from revisions aimed at enhancing clarity and deepening the critical analysis in several sections. Specific comments are as follows.

Answer: Dear reviewer. Thank you for the detailed text and figure analysis. We have tried to make all improvements according to your comments:

  • In the section of “Genetic modifications for Tregs stabilization throw cytokine and cAMP signaling control”, the discussion of the orthogonal IL-2 system and mbIL-2 is excellent. However, for IL-33 and IL-35, the review mostly focuses on the effects of cytokine administration rather than the engineering of Tregs to exploit these pathways, which is the core theme. This section should be reframed to explicitly discuss the potential and challenges of, for example, engineering Tregs to constitutively express ST2 or the IL-35 receptor, and what the expected functional outcomes and potential risks might be.

Answer 1:

We agree with your opinion. The corresponding information  was  added in Sections Interleukin 33 (IL33) signalling and Interleukin 35 (IL35) signalling. More detailed description of IL35 signalling and its interaction with FOXP3 expression and Treg stabilization was added. Results  of tests with genetically modified ST2 upregulated Tregs was added.

  • Certain sections lack sufficient depth, with relatively general descriptions of specific mechanisms, such as the precise role of the IL-35 signaling pathway in Tregs and the direct link between cAMP signaling and FOXP3 expression. These aspects warrant further elaboration. It is recommended to enhance the key mechanistic discussions by incorporating more detailed molecular-level insights, including the interactions among critical signaling proteins and the transcriptional regulatory networks governing FOXP3 expression

Answer 2:

 We agree with your opinion. The corresponding information  was  added in Sections Interleukin 35 (IL35) signalling and cAMP signalling.

  • In the section of “Complex Approach” is a key section but is currently underdeveloped. The proposed pipeline is logical but highly generic. The authors should significantly expand and add specificity to "Complex Approach" with testable hypotheses and a discussion of practical challenges.

How would one prioritize which stabilization method(s) to use? Would a combination be synergistic or redundant?

Discuss the practical and regulatory hurdles of implementing such a multi-layered engineering approach.

"The most secure, stable and effective stabilized autologous CAR stabilized Treg cell line can be selected after a series of tests". What specific in vitro and in vivo assays would be used for this selection? The authors mention stability tests; they should specify metrics.

Answer 3:

Thank you for your remark. The section “Complex Approach” was expanded. The positive and negative sides of synergic and single methods approaches were discussed. The optimal for authors opinion methods for functional tests of modified Tregs  stability and efficiency were described. Practical hurdles of implementing such a multi-layered engineering approach were discussed.

  • Perform a careful proofread of all figures and captions. “Potential side effects of IL2 therapy are illustrated on Fig. 2”, the figure is wrongly matched, this content is displayed in Fig. 1 rather than Fig. 2.

Answer 4: Thank you for your remark. Now the noticed figure number is correct. All figure numbers were checked.

  • Authors are kindly reminded to pay attention to formatting details; for instance, terms such as "in vivo" and "in vitro" should be typeset in italics in accordance with standard scientific writing conventions.

Answer 5: Thank you, that you have noticed that. The corresponding changes were made in text.

Extra answer: The English editing was done.

Reviewer 2 Report

Comments and Suggestions for Authors

This mini-review clarifies a complex strategy that integrates individual enhancement axes—including IL-2/33/35/cAMP signaling, FOXP3 splicing, and epigenetic modification—with CAR-mediated pancreatic islet homing; it further provides a perspective that incorporates the ancillary anti-inflammatory mechanism of ST2-engineered Tregs inducing M2 macrophage polarization into the context of type 1 diabetes, offers a comprehensive review of engineering approaches for Treg-selective IL-2 signaling such as mbIL-2 and oIL2Rβ systems, and proposes a design hypothesis to extrapolate FOXP3 SSO-induced FOXP3FL bias for application in T1D.

Major Comments

  1. Specificity of evidence for T1D: Much of the data is extrapolated from other diseases/models (GVHD, cardiac transplantation, asthma, ALS), and systematic comparative figures/tables for T1D-specific in vivo/clinical data are lacking.
  2. Depth of CAR target summary: Insufficient causal analysis of success and limitations (e.g., tonic signaling, self-recognition) for insulin/GAD65/HPi2/IAg7–B:10-23. Clarification of Table 1 regarding results, study dates, and models is needed.
  3. Manufacturing and regulatory considerations: Requires discussion of GMP manufacturing quality attributes (identity, purity, potency, stability), release testing for migratory capacity/stability, TSDR/histone marker assays, and an overview of regulatory requirements.

Minor Comments

  1. Add columns to Table 1 for mouse strains, human data, and model type (humanized/syngeneic).
  2. Specify FOXP3 SSO sequence design, delivery methods, off-target evaluation, and validation status in human T1D-derived Tregs.
  3. Supplement details on ST2-engineered Tregs: frequency of ST2 expression in human peripheral Tregs, relationship to tissue residency, and cytokine dependency.
  4. Improve resolution and legends of Figures 1–3; fix formatting inconsistencies in Table 2; ensure initial definition of abbreviations (Teff, GVHD, NRG, TSDR, AC-9, etc.); resolve FOXP3/FoxP3 notation inconsistencies; correct spelling/grammar errors (e.g., 'throw' → 'through', 'authologous' → 'autologous', 'monometylation' → 'monomethylation').

Author Response

Answers to reviewer 2 comments:

This mini-review clarifies a complex strategy that integrates individual enhancement axes—including IL-2/33/35/cAMP signaling, FOXP3 splicing, and epigenetic modification—with CAR-mediated pancreatic islet homing; it further provides a perspective that incorporates the ancillary anti-inflammatory mechanism of ST2-engineered Tregs inducing M2 macrophage polarization into the context of type 1 diabetes, offers a comprehensive review of engineering approaches for Treg-selective IL-2 signaling such as mbIL-2 and oIL2Rβ systems, and proposes a design hypothesis to extrapolate FOXP3 SSO-induced FOXP3FL bias for application in T1D.

Answer: Dear reviewer. Thank you for the detailed text and figure analysis. We have tried to make all improvements according to your comments:

Major Comments

  • Specificity of evidence for T1D: Much of the data is extrapolated from other diseases/models (GVHD, cardiac transplantation, asthma, ALS), and systematic comparative figures/tables for T1D-specific in vivo/clinical data are lacking.

We agree with your opinion. Introduction section and table 1 were rewritten and expanded.

Answer 1.1:

Thank you for your comment. Unfortunately, we have not found published data on genetically modified Tregs for the treatment of T1D. We have referenced the available evidence from polyclonal Treg trials in our introduction. The discussion of IL-2, IL-35, and IL-33/ST2 pathways as potential genetic modification targets is based on data from studies involving their exogenous modulation in the context of T1D, which is the focus of Table 2.

  1. Depth of CAR target summary: Insufficient causal analysis of success and limitations (e.g., tonic signaling, self-recognition) for insulin/GAD65/HPi2/IAg7–B:10-23. Clarification of Table 1 regarding results, study dates, and models is needed.

Answer 1.2:

We are grateful for this feedback. In response, we have expanded the analysis of the successes and limitations of CAR-Treg therapy in the "Introduction" section. Furthermore, we have supplemented the relevant table with additional data regarding the animal models, cell lines, and experimental timeframes.

  1. Manufacturing and regulatory considerations: Requires discussion of GMP manufacturing quality attributes (identity, purity, potency, stability), release testing for migratory capacity/stability, TSDR/histone marker assays, and an overview of regulatory requirements.

Answer 1.3:

Thank you for this suggestion. We have added a "The Application of Good Manufacturing Practice (GMP) in Genetically Modified Treg Therapy" section as suggested.

Minor Comments

  1. Add columns to Table 1 for mouse strains, human data, and model type (humanized/syngeneic).

Answer 2.1:

Thank you for your remark. The table 1 was expanded, information about strains and T1D model type was added.

  1. Specify FOXP3 SSO sequence design, delivery methods, off-target evaluation, and validation status in human T1D-derived Tregs.

Answer 2.2:

 We agree with your opinion. The corresponding information were added in Section Modulation of Treg stability, suppressive and proliferative activity through FOXP3 alternative splicing.

We have addressed these concerns in the revised manuscript.

  1. Supplement details on ST2-engineered Tregs: frequency of ST2 expression in human peripheral Tregs, relationship to tissue residency, and cytokine dependency (

Лена).

Answer 2.3:

Thank you for your comment.  The suggested points were incorporated into the revised manuscript.

  1. Improve resolution and legends of Figures 1–3; fix formatting inconsistencies in Table 2; ensure initial definition of abbreviations (Teff, GVHD, NRG, TSDR, AC-9, etc.); resolve FOXP3/FoxP3 notation inconsistencies; correct spelling/grammar errors (e.g., 'throw' → 'through', 'authologous' → 'autologous', 'monometylation' → 'monomethylation').

Answer 2.4:

 Thank you for your remark. The corresponding errors were corrected. The resolution of the figures was upscaled to 500PPI. All initial abbreviations in text were checked.

Extra answer:

The English editing was done.

Round 2

Reviewer 1 Report

Comments and Suggestions for Authors

All comments have been addressed, suggest to accept for publication

Reviewer 2 Report

Comments and Suggestions for Authors

Thank you for the authors' precise and thoughtful responses to the review comments. Their careful revisions have significantly improved the clarity and quality of the manuscript, and I appreciate your diligence in addressing each point thoroughly.